# Invertible DenseNets with Concatenated LipSwish

**Yura Perugachi-Diaz**
Vrije Universiteit Amsterdam
y.m.perugachidiaz@vu.nl

**Jakub M. Tomczak**
Vrije Universiteit Amsterdam
j.m.tomczak@vu.nl

**Sandjai Bhulai**
Vrije Universiteit Amsterdam
s.bhulai@vu.nl

## Abstract

We introduce Invertible Dense Networks (i-DenseNets), a more parameter efficient extension of Residual Flows. The method relies on an analysis of the Lipschitz continuity of the concatenation in DenseNets, where we enforce invertibility of the network by satisfying the Lipschitz constant. Furthermore, we propose a learnable weighted concatenation, which not only improves the model performance but also indicates the importance of the concatenated weighted representation. Additionally, we introduce the Concatenated LipSwish as activation function, for which we show how to enforce the Lipschitz condition and which boosts performance. The new architecture, i-DenseNet, out-performs Residual Flow and other flow-based models on density estimation evaluated in bits per dimension, where we utilize an equal parameter budget. Moreover, we show that the proposed model out-performs Residual Flows when trained as a hybrid model where the model is both a generative and a discriminative model.

## 1   Introduction

Neural networks are widely used to parameterize non-linear models in supervised learning tasks such as classification. In addition, they are also utilized to build flexible density estimators of the true distribution of the observed data [25, 33]. The resulting deep density estimators, also called deep generative models, can be further used to generate realistic-looking images that are hard to separate from real ones, detection of adversarial attacks [9, 17], and for hybrid modeling [27] which have the property to both predict a label (classify) and generate.

Many deep generative models are trained by maximizing the (log-)likelihood function and their architectures come in different designs. For instance, causal convolutional neural networks are used to parameterize autoregressive models [28, 29] or various neural networks can be utilized in Variational Auto-Encoders [19, 32]. The other group of likelihood-based deep density estimators, *flow-based models* (or *flows*), consist of invertible neural networks since they are used to compute the likelihood through the change of variable formula [31, 37, 36]. The main difference that determines an exact computation or approximation of the likelihood function for a flow-based model lies in the design of the transformation layer and tractability of the Jacobian-determinant. Many flow-based models formulate the transformation that is invertible and its Jacobian is tractable [3, 6–8, 21, 30, 31, 38].

Recently, Behrmann et al. [2] proposed a different approach, namely, deep-residual blocks as a transformation layer. The deep-residual networks (ResNets) of [12] are known for their successes in supervised learning approaches. In a ResNet block, each input of the block is added to the output, which forms the input for the next block. Since ResNets are not necessarily invertible, Behrmann et al. [2] enforce the Lipschitz constant of the transformation to be smaller than 1 (i.e., it becomes a contraction) that allows applying an iterative procedure to invert the network. Furthermore, Chen et al. [4] proposed Residual Flows, an improvement of i-ResNets, that uses an unbiased estimator for the logarithm of the Jacobian-determinant.

35th Conference on Neural Information Processing Systems (NeurIPS 2021)

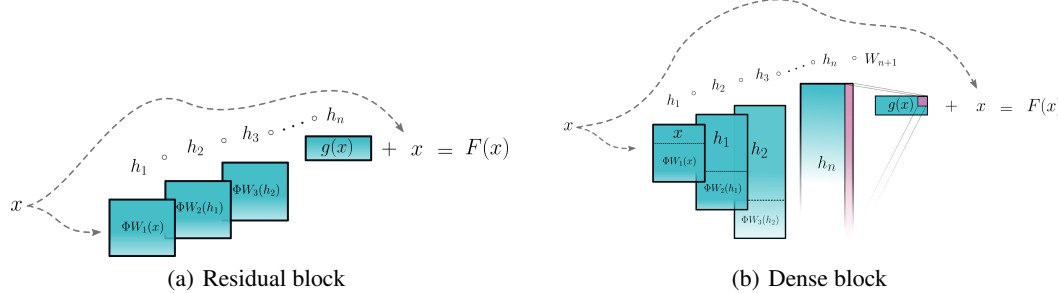

(a) Residual block             (b) Dense block

Figure 1: A schematic representation for: (a) a residual block, (b) a dense block. The pink part in (b) expresses a $1 \times 1$ convolution to reduce the dimension of the last dense layer. $W_i$ denotes the (convolutional) layer at step $i$ that satisfy $||W_i||_2 < 1$.

In supervised learning, an architecture that uses fewer parameters and is even more powerful than the deep-residual network is the Densely Connected Convolution Network (DenseNet), which was first presented in [15]. Contrary to a ResNet block, a DenseNet layer consists of a concatenation of the input with the output. The network showed to improve significantly in recognition tasks on benchmark datasets such as CIFAR10, SVHN, and ImageNet, by using fewer computations and having fewer parameters than ResNets while performing at a similar level.

In this work, we extend Residual Flows [2, 4], and use densely connected blocks (DenseBlocks) as a residual layer. First, we introduce invertible Dense Networks (i-DenseNets), and we show that we can derive a bound on the Lipschitz constant to create an invertible flow-based model. Furthermore, we propose the Concatenated LipSwish (CLipSwish) as an activation function, and derive a stronger Lipschitz bound. The CLipSwish function preserves more signal than LipSwish activation functions. Finally, we demonstrate how i-DenseNets can be efficiently trained as a generative model, outperforming Residual Flows and other flow-based models under an equal parameter budget.

## 2   Background

**Flow-based models**     Let us consider a vector of observable variables $x \in \mathbb{R}^d$ and a vector of latent variables $z \in \mathbb{R}^d$. We define a bijective function $f : \mathbb{R}^d \to \mathbb{R}^d$ that maps a latent variable to a datapoint $x = f(z)$. Since $f$ is invertible, we define its inverse as $F = f^{-1}$. We use the *change of variables formula* to compute the likelihood of a datapoint $x$ after taking the logarithm, that is:

$$\ln p_X(x) = \ln p_Z(z) + \ln |\det J_F(x)|, \tag{1}$$

where $p_Z(z)$ is a base distribution (e.g., the standard Gaussian) and $J_F(x)$ is the Jacobian of $F$ at $x$. The bijective transformation is typically constructed as a sequence of $K$ invertible transformations, $x = f_K \circ \cdots \circ f_1(z)$, and a single transformation $f_k$ is referred to as a *flow* [31]. The change of variables formula allows evaluating the data in a tractable manner. Moreover, the flows are trained using the log-likelihood objective where the Jacobian-determinant compensates the change of volume of the invertible transformations.

**Residual flows**     Behrmann et al. [2] construct an invertible ResNet layer which is only constrained in Lipschitz continuity. A ResNet is defined as: $F(x) = x + g(x)$, where $g$ is modeled by a (convolutional) neural network and $F$ represents a ResNet layer (see Figure 1(a)) which is in general not invertible. However, $g$ is constructed in such a way that it satisfies the Lipschitz constant being strictly lower than 1, $\mathrm{Lip}(g) < 1$, by using spectral normalization of [10, 26]:

$$\mathrm{Lip}(g) < 1, \quad \text{if} \quad ||W_i||_2 < 1, \tag{2}$$

where $|| \cdot ||_2$ is the $\ell_2$ matrix norm. Then $\mathrm{Lip}(g) = \mathrm{K} < 1$ and $\mathrm{Lip}(F) < 1 + \mathrm{K}$. Only in this specific case the Banach fixed-point theorem holds and ResNet layer $F$ has a unique inverse. As a result, the inverse can be approximated by fixed-point iterations.

To estimate the log-determinant is, especially for high-dimensional spaces, computationally intractable due to expensive computations. Since ResNet blocks have a constrained Lipschitz constant,

the log-likelihood estimation of Equation (1) can be transformed to a version where the logarithm of the Jacobian-determinant is cheaper to compute, tractable, and approximated with guaranteed convergence [2]:

$$\ln p(x) = \ln p(f(x)) + \text{tr}\left(\sum_{k=1}^{\infty} \frac{(-1)^{k+1}}{k}[J_g(x)]^k\right),\tag{3}$$

where $J_g(x)$ is the Jacobian of $g$ at $x$ that satisfies $||J_g||_2 < 1$. The Skilling-Hutchinson trace estimator [35, 16] is used to compute the trace at a lower cost than to fully compute the trace of the Jacobian. Residual Flows [4] use an improved method to estimate the power series at an even lower cost with an unbiased estimator based on "Russian roulette" of [18]. Intuitively, the method estimates the infinite sum of the power series by evaluating a finite amount of terms. In return, this leads to less computation of terms compared to invertible residual networks. To avoid derivative saturation, which occurs when the second derivative is zero in large regions, the LipSwish activation is proposed.

## 3 Invertible Dense Networks

In this section, we propose Invertible Dense Networks by using a DenseBlock as a residual layer. We show how the network can be parameterized as a flow-based model and refer to the resulting model as i-DenseNets. The code can be retrieved from: https://github.com/yperugachidiaz/invertible_densenets.

### 3.1 Dense blocks

The main component of the proposed flow-based model is a DenseBlock that is defined as a function $F : \mathbb{R}^d \to \mathbb{R}^d$ with $F(x) = x + g(x)$, where $g$ consists of dense layers $\{h_i\}_{i=1}^n$. Note that an important modification to make the model invertible is to output $x + g(x)$ whereas a standard DenseBlock would only output $g(x)$. The function $g$ is expressed as follows:

$$g(x) = W_{n+1} \circ h_n \circ \cdots \circ h_1(x),\tag{4}$$

where $W_{n+1}$ represents a $1 \times 1$ convolution to match the output size of $\mathbb{R}^d$. A layer $h_i$ consists of two parts concatenated to each other. The upper part is a copy of the input signal. The lower part consists of the transformed input, where the transformation is a multiplication of (convolutional) weights $W_i$ with the input signal, followed by a non-linearity $\phi$ having $\text{Lip}(\phi) \leq 1$, such as ReLU, ELU, LipSwish, or tanh. As an example, a dense layer $h_2$ can be composed as follows:

$$h_1(x) = \begin{bmatrix} x \\ \phi(W_1 x) \end{bmatrix}, \; h_2(h_1(x)) = \begin{bmatrix} h_1(x) \\ \phi(W_2 h_1(x)) \end{bmatrix}.\tag{5}$$

In Figure 1, we schematically outline a residual block (Figure 1(a)) and a dense block (Figure 1(b)). We refer to concatenation *depth* as the number of dense layers in a DenseBlock and *growth* as the channel growth size of the transformation in the lower part.

### 3.2 Constraining the Lipschitz constant

If we enforce function $g$ to satisfy $\text{Lip}(g) < 1$, then DenseBlock $F$ is invertible since the Banach fixed point theorem holds. As a result, the inverse can be approximated in the same manner as in [2]. To satisfy $\text{Lip}(g) < 1$, we need to enforce $\text{Lip}(h_i) < 1$ for all $n$ layers, since $\text{Lip}(g) \leq \text{Lip}(h_{n+1}) \cdot \ldots \cdot \text{Lip}(h_1)$. Therefore, we first need to determine the Lipschitz constant for a dense layer $h_i$. For the full derivation, see Appendix A. We know that a function $f$ is K-Lipschitz if for all points $v$ and $w$ the following holds :

$$d_Y(f(v), f(w)) \leq \text{K} d_X(v, w),\tag{6}$$

where we assume that the distance metrics $d_X = d_Y = d$ are chosen to be the $\ell_2$-norm. Further, let two functions $f_1$ and $f_2$ be concatenated in $h$:

$$h_v = \begin{bmatrix} f_1(v) \\ f_2(v) \end{bmatrix}, \quad h_w = \begin{bmatrix} f_1(w) \\ f_2(w) \end{bmatrix},\tag{7}$$

where function $f_1$ is the upper part and $f_2$ is the lower part. We can now find an analytical form to express a limit on K for the dense layer in the form of Equation (6):

$$d(h_v, h_w)^2 = d(f_1(v), f_1(w))^2 + d(f_2(v), f_2(w))^2,$$
$$d(h_v, h_w)^2 \leq (K_1^2 + K_2^2)d(v, w)^2, \tag{8}$$

where we know that the Lipschitz constant of $h$ consist of two parts, namely, $\mathrm{Lip}(f_1) = K_1$ and $\mathrm{Lip}(f_2) = K_2$. Therefore, the Lipschitz constant of layer $h$ can be expressed as:

$$\mathrm{Lip}(h) = \sqrt{(K_1^2 + K_2^2)}. \tag{9}$$

With spectral normalization of Equation (2), we know that we can enforce (convolutional) weights $W_i$ to be at most 1-Lipschitz. Hence, for all $n$ dense layers we apply the spectral normalization on the lower part which locally enforces $\mathrm{Lip}(f_2) = K_2 < 1$. Further, since we enforce each layer $h_i$ to be at most 1-Lipschitz and we start with $h_1$, where $f_1(x) = x$, we know that $\mathrm{Lip}(f_1) = 1$. Therefore, the Lipschitz constant of an entire layer can be at most $\mathrm{Lip}(h) < \sqrt{1^2 + 1^2} = \sqrt{2}$, thus dividing by this limit enforces each layer to be at most 1-Lipschitz.

### 3.3 Learnable weighted concatenation

We have shown that we can enforce an entire dense layer to have $\mathrm{Lip}(h_i) < 1$ by applying a spectral norm on the (convolutional) weights $W_i$ and then divide the layer $h_i$ by $\sqrt{2}$. Although learning a weighting between the upper and lower part would barely affect a standard dense layer, it matters in this case because the layers are regularized to be 1-Lipschitz. To optimize and learn the importance of the concatenated representations, we introduce learnable parameters $\eta_1$ and $\eta_2$ for, respectively, the upper and lower part of each layer $h_i$.

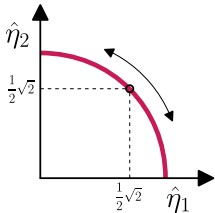

Figure 2: Range of the possible normalized parameters $\hat{\eta}_1$ and $\hat{\eta}_2$.

Since the upper and lower part of the layer can be at most 1-Lipschitz, multiplication by these factors results in functions that are at most $\eta_1$-Lipschitz and $\eta_2$-Lipschitz. As indicated by Equation (9), the layer is then at most $\sqrt{\eta_1^2 + \eta_2^2}$−Lipschitz. Dividing by this factor results in a bound that is at most 1-Lipschitz.

In practice, we initialize $\eta_1$ and $\eta_2$ at value 1 and during training use a softplus function to avoid them being negative. The range of the normalized parameters is between $\hat{\eta}_1, \hat{\eta}_2 \in [0, 1]$ and can be expressed on the unit circle as shown in Figure 2. In the special case where $\eta_1 = \eta_2$, the normalized parameters are $\hat{\eta}_1 = \hat{\eta}_2 = \frac{1}{2}\sqrt{2}$. This case corresponds to the situation in Section 3.2 where the concatenation is not learned. An additional advantage is that the normalized $\hat{\eta}_1$ and $\hat{\eta}_2$ express the importance of the upper and lower signal. For example, when $\hat{\eta}_1 > \hat{\eta}_2$, the input signal is of more importance than the transformed signal.

### 3.4 CLipSwish

When a deep neural network is bounded to be 1-Lipschitz, in practice each consecutive layer reduces the Jacobian-norm. As a result, the Jacobian-norm of the entire network is becoming much smaller than 1 and the expressive power is getting lost. This is known as *gradient norm attenuation* [1, 24]. This problem arises in activation functions in regions where the derivative is small, such as the left tail of the ReLU and the LipSwish. Non-linearities $\phi$ modeled in i-DenseNets are required to be at most 1-Lipschitz and, thus, face gradient-norm attenuation issues. For this reason we introduce a new activation function, which mitigates these issues.

Recall that Residual Flows use the LipSwish activation function [4]:

$$\mathrm{LipSwish}(x) = x\sigma(\beta x)/1.1, \tag{10}$$

where $\sigma(\beta x) = 1/(1 + \exp(-x\beta))$ is the sigmoid , $\beta$ is a learnable constant, initialized at 0.5 and is passed through a softplus to be strictly positive. This activation function is not only $\mathrm{Lip}(\mathrm{LipSwish}) = 1$ but also resolves the derivative saturation problem [4]. However, the LipSwish function has large ranges on the negative axis where its derivative is close to zero.

Therefore, we propose the Concatenated LipSwish (CLipSwish) which concatenates two LipSwish functions with inputs $x$ and $-x$. This is a concatenated activation function as in [34] but using a LipSwish instead of a ReLU. Intuitively, even if an input lies in the tail of the upper part, it will have a larger derivative in the bottom part and thus suffer less from gradient norm attenuation. Since using CLipSwish increases the channel growth and to stay inline with the channel growth that non-concatenated activation functions use, we use a lower channel growth when using CLipSwish. To utilize the CLipSwish, we need to derive Lipschitz continuity of the activation function $\Phi$ defined below and enforce it to be 1-Lipschitz. We could use the result obtained in Equation (9) to obtain a $\sqrt{2}$-bound, however, by using knowledge about the activation function $\Phi$, we can derive a tighter $1.004 < \sqrt{2}$ bound. In general, a tighter bound is preferred since more expressive power will be preserved in the network. To start with, we define function $\Phi : \mathbb{R} \to \mathbb{R}^2$ for a point $x$ as:

$$\Phi(x) = \begin{bmatrix} \phi_1(x) \\ \phi_2(x) \end{bmatrix} = \begin{bmatrix} \text{LipSwish}(x) \\ \text{LipSwish}(-x) \end{bmatrix}, \quad \text{CLipSwish(x)} = \Phi(\text{x})/\text{Lip}(\Phi), \tag{11}$$

where the LipSwish is given by Equation (10) and the derivative of $\Phi(x)$ exists. To find $\text{Lip}(\Phi)$ we use that for a differentiable $\ell_2$-Lipschitz bounded function $\Phi$, the following identity holds:

$$\text{Lip}(\Phi) = \sup_x ||J_\Phi(x)||_2, \tag{12}$$

where $J_\Phi(x)$ is the Jacobian of $\Phi$ at $x$ and $||\cdot||_2$ represents the induced matrix norm which is equal to the spectral norm of the matrix. Rewriting the spectral norm results in solving: $\det(J_\Phi(x)^T J_\Phi(x) - \lambda I_n) = 0$, which gives us the final result (see Appendix A.3.1 for the full derivation):

$$\sup_x ||J_\Phi(x)||_2 = \sup_x \sigma_{\max}(J_\Phi(x)) = \sup_x \sqrt{\left(\frac{\partial \phi_1(x)}{\partial x}\right)^2 + \left(\frac{\partial \phi_2(x)}{\partial x}\right)^2}, \tag{13}$$

where $\sigma_{\max}(\cdot)$ is the largest singular value. Now $\text{Lip}(\Phi)$ is the upper bound of the CLipSwish and is equal to the supremum of: $\text{Lip}(\Phi) = \sup_x ||J_\Phi(x)||_2 \approx 1.004$, for all values of $\beta$. This can be numerically computed by any solver, by determining the extreme values of Equation (13). Therefore, dividing $\Phi(x)$ by its upper bound 1.004 results in $\text{Lip}(\text{CLipSwish}) = 1$. The generalization to higher dimensions can be found in Appendix A.3.2. The analysis of preservation of signals for (CLip)Swish activation by simulations can be found in Section 5.1.

## 4   Experiments

To make a clear comparison between the performance of Residual Flows and i-DenseNets, we train both models on 2-dimensional toy data and high-dimensional image data: CIFAR10 [22] and ImageNet32 [5]. Since we have a constrained computational budget, we use smaller architectures for the exploration of the network architectures. An in-depth analysis of different settings and experiments can be found in Section 5. For density estimation, we run the full model with the best settings for 1,000 epochs on CIFAR10 and 20 epochs on ImageNet32 where we use single-seed results following [2, 4, 20], due to little fluctuations in performance. In all cases, we use the density estimation results of the Residual Flow and other flow-based models using uniform dequantization to create a fair comparison and benchmark these with i-DenseNets. We train i-DenseNets with learnable weighted concate-

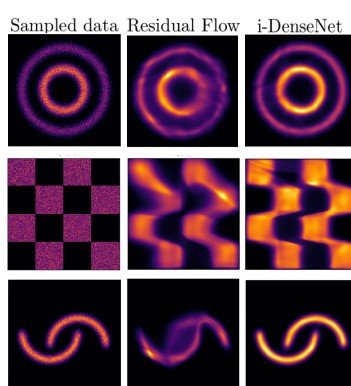

Figure 3: Density estimation for **smaller** architectures of Residual Flows and i-DenseNets, trained on 2-dimensional toy data.

nation (LC) and CLipSwish as the activation function, and utilize a similar number of parameters for i-DenseNets as Residual Flows; this can be found in Table 2. i-DenseNets uses slightly fewer parameters than the Residual Flow. A detailed description of the architectures can be found in Appendix B. To speed up training, we use 4 GPUs.

### 4.1 Toy data

We start with testing i-DenseNets and Residual Flows on toy data, where we use smaller architectures. Instead of 100 flow blocks, we use 10 flow blocks. We train both models for 50,000 iterations and, at the end of the training, we visualize the learned distributions.

The results of the learned density distributions are presented in Figure 3. We observe that Residual Flows are capable to capture high-probability areas. However, they have trouble with learning low probability regions for two circles and moons. i-DenseNets are capable of capturing all regions of the datasets. The good performance of i-DenseNets is also reflected in better performance in terms of the negative-log-likelihood (see Table 1).

Table 1: Negative log-likelihood results on test data in nats (toy data). i-DenseNets w/ and w/o LC are compared with the Residual Flow.

| Model | 2 circles | checkerboard | 2 moons |
|---|---|---|---|
| Residual Flows | 3.44 | 3.81 | 2.60 |
| i-DenseNets | 3.32 | 3.68 | **2.39** |
| **i-DenseNets+LC** | **3.30** | **3.66** | **2.39** |

### 4.2 Density Estimation

We test the full i-DenseNet models with LC and CLipSwish activation. To utilize a similar number of parameters as the Residual Flow with 3 scale levels and flow blocks set to 16 per scale trained on CIFAR10, we set for the same number of blocks, DenseNets growth to 172 with a depth of 3. Residual Flow trained on ImageNet32 uses 3 scale levels with 32 flow blocks per scale, therefore, we set for the same number of blocks DenseNets growth to 172 and depth of 3 to utilize a similar number of parameters. DenseNets depth set to 3 proved to be the best settings for smaller architectures; see the analysis in Section 5.

Table 2: The number of parameters of Residual Flows and i-DenseNets for the full models as trained in Chen et al. [4]. In brackets, the number of parameters of the smaller models.

| Model/Data | CIFAR10 | ImageNet32 |
|---|---|---|
| Residual Flows | 25.2M (8.7M) | 47.1M |
| i-DenseNets | 24.9M (8.7M) | 47.0M |

The density estimation on CIFAR10 and ImageNet32 are benchmarked against the results of Residual Flows and other comparable flow-based models, where the results are retrieved from Chen et al. [4]. We measure performances in bits per dimension (bpd). The results can be found in Table 3. We find that i-DenseNets out-perform Residual Flows and other comparable flow-based models on all considered datasets in terms of bpd. On CIFAR10, i-DenseNet achieves 3.25bpd, against 3.28bpd of the Residual Flow. On ImageNet32 i-DenseNet achieves 3.98bpd against 4.01bpd of the Residual Flow. Samples of the i-DenseNet models can be found in Figure 4. Samples of the model trained on CIFAR10 are presented in Figure 4(b) and samples of the model trained on ImageNet32

Table 3: Density estimation results in bits per dimension for models using **uniform dequantization**. In brackets results for the smaller Residual Flow and i-DenseNet run for 200 epochs.

| Model | CIFAR10 | ImageNet32 |
|---|---|---|
| Real NVP [8] | 3.49 | 4.28 |
| Glow [20] | 3.35 | 4.09 |
| FFJORD [11] | 3.40 | - |
| Flow++ [13] | 3.29 | - |
| ConvSNF [14] | 3.29 | - |
| i-ResNet [2] | 3.45 | - |
| Residual Flow [4] | 3.28 (3.42) | 4.01 |
| **i-DenseNet** | **3.25** (3.37) | **3.98** |

in Figure 4(d). For more unconditional samples, see Appendix C.1. Note that this work does not compare against flow-based models using variational dequantization. Instead we focus on extending and making a fair comparison with Residual Flows which, similar to other flow-based models, use uniform dequantization. For reference note that Flow++ [13] with variational dequantization obtains 3.08bpd on CIFAR10 and 3.86bpd on ImageNet32 that is better than the model with uniform dequantization which achieves 3.29bpd on CIFAR10.

### 4.3 Hybrid Modeling

Besides density estimation, we also experiment with hybrid modeling [27]. We train the joint distribution $p(\mathbf{x}, y) = p(\mathbf{x}) \, p(y|\mathbf{x})$, where $p(\mathbf{x})$ is modeled with a generative model and $p(y|\mathbf{x})$ is modeled with a classifier, which uses the features of the transformed image onto the latent space.

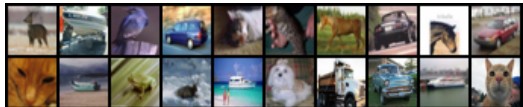 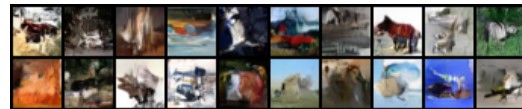

(a) Real CIFAR10 images.    (b) Samples of i-DenseNets trained on CIFAR10.

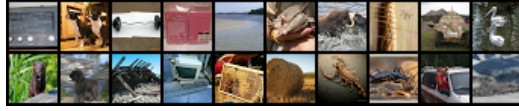

(c) Real ImageNet32 images.    (d) Samples of i-DenseNets trained on ImageNet32.

Figure 4: Real and samples of CIFAR10 and ImageNet32 data

Due to the different dimensionalities of $y$ and $\mathbf{x}$, the emphasis of the likelihood objective is more likely to be focused on $p(\mathbf{x})$ and a scaling factor for a weighted maximum likelihood objective is suggested, $\mathbb{E}_{x,y\sim\mathcal{D}}\left[\log p(y|\mathbf{x}) + \lambda \log p(\mathbf{x})\right]$, where $\lambda$ is the scaling factor expressing the trade-off between the generative and discriminative parts. Unlike [27] where a linear layer is integrated on top of the latent representation, we use the architecture of [4] where the set of features are obtained after every scale level. Then, they are concatenated and are followed by a linear softmax classifier. We compare our experiments with the results of [4] where Residual Flow, coupling blocks [7] and $1 \times 1$ convolutions [20] are evaluated.

Table 4 presents the hybrid modeling results on CIFAR10 where we used $\lambda = \{0, \frac{1}{D}, 1\}$. We run the three models for 400 epochs and note that the model with $\lambda = 1$ was not fully converged in both accuracy and bits per dimension after training. The classifier model obtains a converged accuracy after around 250 epochs. This is in line with the accuracy for the model with $\lambda = \frac{1}{D}$, yet based on bits per dimension the model was not fully

Table 4: Results of hybrid modeling on CIFAR10. Arrows indicate if low or high values are of importance. Results are average over the last 5 epochs.

| | $\lambda = 0$ | $\lambda = \frac{1}{D}$ | | $\lambda = 1$ | |
|---|---|---|---|---|---|
| Model \Evaluation | Acc ↑ | Acc ↑ | bpd ↓ | Acc ↑ | bpd ↓ |
| Coupling | 89.77% | 87.58% | 4.30 | 67.62% | 3.54 |
| + $1 \times 1$ conv | 90.82% | 87.96% | 4.09 | 67.38% | 3.47 |
| Residual Blocks (full) | 91.78% | 90.47% | 3.62 | 70.32% | 3.39 |
| Dense Blocks (full) | **92.40%** | **90.79%** | **3.49** | **75.67%** | **3.31** |

converged after 400 epochs. This indicates that even though the accuracy is not further improved, the model keeps optimizing the bits per dimension which gives room for future research. Results in Table 4 show the average result over the last 5 epochs. We find that Dense Blocks out-perform Residual Blocks for all possible $\lambda$ settings. Interestingly, Dense Blocks have the biggest impact using no penalty ($\lambda = 1$) compared to the other models. We obtain an accuracy of 75.67% with 3.31bpd, compared to 70.32% accuracy and 3.39bpd of Residual Blocks, indicating that Dense Blocks significantly improves classification performance with more than 5%. In general, the Dense Block hybrid model is out-performing Real NVP, Glow, FFJORD, and i-ResNet in bits per dimension (see Appendix C.2 for samples of the hybrid models).

## 5 Analysis and future work

To get a better understanding of i-DenseNets, we perform additional experiments, explore different settings, analyze the results of the model and discuss future work. We use smaller architectures for these experiments due to a limited computational budget. For Residual Flows and i-DenseNets we use 3 scale levels set to 4 Flow blocks instead of 16 per scale level and train models on CIFAR10 for 200 epochs. We will start with a short explanation of the limitations of 1-Lipschitz deep neural networks.

### 5.1 Analysis of activations and preservation of signals

Since gradient-norm attenuation can arise in 1-Lipschitz bounded deep neural nets, we analyze how much signal of activation functions is preserved by examining the maximum and average distance ratios of sigmoid, LipSwish, and CLipSwish. Note that the maximum distance ratio approaches the Lipschitz constant and it is desired that the average distance ratio remains high.

We sample 100,000 datapoints $v, w \sim \mathcal{N}(0, 1)$ with dimension set to $D = \{1, 128, 1024\}$. We compute the mean and maximum of the sampled ratios with: $\ell_2(\phi(v), \phi(w))/\ell_2(v, w)$ and analyze the expressive power of each function. Table 5 shows the results. We find that CLipSwish for all dimensions preserves most of the signal on average compared to the other non-linearities. This may explain why

Table 5: The mean and maximum ratio for different dimensions with sample size set to 100,000.

| Activation\Measure | $D = 1$ | | $D = 128$ | | $D = 1024$ | |
|---|---|---|---|---|---|---|
| | Mean | Max | Mean | Max | Mean | Max |
| Sigmoid | 0.22 | 0.25 | 0.21 | 0.22 | 0.21 | 0.21 |
| LipSwish | 0.46 | 1.0 | 0.51 | 0.64 | 0.51 | 0.55 |
| CLipSwish | 0.72 | 1.0 | 0.71 | 0.77 | 0.71 | 0.73 |
| Identity | 1.0 | 1.0 | 1.0 | 1.0 | 1.0 | 1.0 |

i-DenseNets with CLipSwish activation achieves better results than using, e.g., LipSwish. This experiment indicates that on randomly sampled points, CLipswish functions suffer from considerably less gradient norm attenuation. Note that sampling from a distribution with larger parameter values is even more pronounced in preference of CLipSwish, see Appendix D.1.

## 5.2 Activation Functions

We start with exploring different activation functions for both networks and test these with the smaller architectures. We compare our CLipSwish to the LipSwish and the LeakyLSwish as an additional baseline, which allows freedom of movement in the left tail as opposed to a standard LipSwish:

$$\text{LeakyLSwish}(x) = \alpha x + (1 - \alpha)\text{LipSwish}(x), \tag{14}$$

with $\alpha \in (0, 1)$ by passing it through a sigmoid function $\sigma$. Here $\alpha$ is a learnable parameter which is initialized at $\alpha = \sigma(-3)$ to mimic the LipSwish at initialization. Note that the dimension for Residual Flows with CLipSwish activation function is set to 652 instead of 512 to maintain a similar number of parameters (8.7M) as with LipSwish activation.

Table 6 shows the results of each model using different activation functions. With 3.37bpd we conclude that i-DenseNet with our CLipSwish as the activation function obtains the best performance compared to the other activation functions, LipSwish and LeakyLSwish. Furthermore, all i-

Table 6: Results in bits per dimensions for small architectures, testing different activation functions.

| Model | LipSwish | LeakyLSwish | CLipSwish |
|---|---|---|---|
| Residual Flow | 3.42 | 3.42 | 3.38 |
| **i-DenseNet** | **3.39** | **3.39** | **3.37** |

DenseNets out-perform Residual Flows with the same activation function. We want to point out that CLipSwish as the activation function not only boosts performance of i-DenseNets but it also significantly improves the performance of Residual Flows with 3.38bpd. The running time for the forward pass, train time and sampling time, expressed in percentage faster or slower than Residual Flow with the same activation functions, can be found in Appendix D.2.

## 5.3 DenseNets concatenation depth

Next, we examine the effect of different concatenation depth settings for i-DenseNets. We run experiments with concatenation depth set to 2, 3, 4, and 5 with CLipSwish. Furthermore, to utilize 8.7M parameters of the Residual Flow, we choose a fixed depth and appropriate DenseNet growth size to have a similar number of parameters. This results in a DenseNet depth 2 with growth size 318 (8.8M), depth 3 with growth 178 (8.7M), depth 4 with growth 122 (8.7M), and depth 5 with growth 92 (8.8M). The effect of each architecture can be found in Figure 5. We observe that the model with a depth of 3 obtains the best scores and after 200 epochs it achieves the lowest bits per dimension with 3.37bpd. A concatenation depth of 5 results in

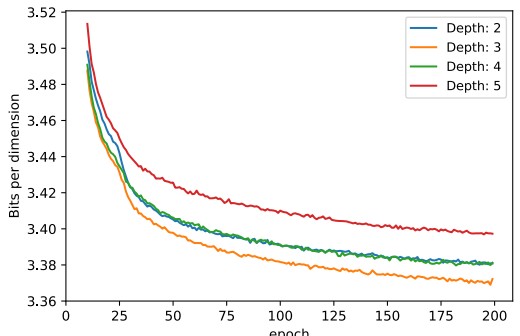

Figure 5: Effect of different concatenation depths with CLipSwish activation function for i-DenseNets in bits per dimension.

3.42bpd after 200 epochs, which is the least preferred. This could indicate that the corresponding DenseNet growth of 92 is too little to capture the density structure sufficiently and due to the deeper depth the network might lose important signals. Further, the figure clearly shows how learnable weighted concatenation after 25 epochs boosts training for all i-DenseNets. See Appendix D.3 for an in-depth analysis of results of the learnable weighted concatenation. Furthermore, we performed an additional experiment (see Appendix D.4) where we extended the width and depth of the ResNet connections in $g(x)$ of Residual Flows in such a way that it matches the size of the i-DenseNet. As a result on CIFAR10 this puts the extended Residual Flow at a considerable advantage as it utilizes 19.1M parameters instead of 8.7M. However, when looking at performance the model performs worse (7.02bpd) than i-Densenets (3.39bpd) and even worse than its original version (3.42bpd) in terms of bpd. A possible explanation of this phenomenon is that by forcing more convolutional layers to be 1-Lipschitz, the gradient norm attenuation problems increase and in practice they become considerably less expressive. This indicates that modeling a DenseNet in $g(x)$ is indeed an important difference that gives better performance.

## 5.4  Future Work

We introduced a new framework, i-DenseNet, that is inspired by Residual Flows and i-ResNets. We demonstrated how i-DenseNets out-performs Residual Flows and alternative flow-based models for density estimation and hybrid modeling, constraint by using uniform dequantization. For future work, we want to address several interesting aspects we came across and where i-DenseNets may be further deployed and explored.

First of all, we find that smaller architectures have more impact on performance than full models compared to Residual Flows. Especially for exploration of the network, we recommend experimenting with smaller architectures or when a limited computational budget is available. This brings us to the second point. Due to a limited budget, we trained and tested i-DenseNets on $32 \times 32$ CIFAR10 and ImageNet32 data. It will be interesting to test higher resolution and other types of datasets.

Further exploration of DenseNets depth and growth for other or higher resolution datasets may be worthwhile. In our studies, deeper DenseNets did not result in better performance. However, it would also be beneficial to further examine the optimization of DenseNets architectures. Similarly, we showed how to constrain DenseBlocks for the $\ell_2$-norm. For future work, it may be interesting to generalize the method to different norm types, as well as the norm for CLipSwish activation function. Note that CLipSwish as activation function not only boosts performance of i-DenseNets but also for Residual Flows. We recommend this activation function for future work.

We want to stress that we focused on extending Residual Flows, which uses uniform dequantization. However, we believe that the performance of our network may be improved using variational dequantization or augmentation. Finally, we found that especially hybrid model with $\lambda = 1$, achieve better performance than its predecessors. This may be worthwhile to further investigate in the future.

**Societal Impact**    We discussed methods to improve normalizing flow, a method that learns high-dimensional distributions. We generated realistic-looking images and also used hybrid models that both predict the label of an image and generate new ones. Besides generating images, these models can be deployed to, e.g., detect adversarial attacks. Additionally, this method is applicable to all different kind of fields such as chemistry or physics. An increasing concern is that generative models in general, have an impact on society. They can not only be used to aid society but can also be used to generate misleading information by those who use these models. Examples of these cases could be generating real-looking documents, Deepfakes or even detection of fraud with the wrong intentions. Even though current flow-based models are not there yet to generate flawless reproductions, this concern should be kept in mind. It even raises the question if these models should be used in practice when detection of misleading information becomes difficult or even impossible to track.

# 6    Conclusion

In this paper, we proposed i-DenseNets, a parameter-efficient alternative to Residual Flows. Our method enforces invertibility by satisfying the Lipschitz continuity in dense layers. In addition, we introduced a version where the concatenation of features is learned during training that indicates which representations are of importance for the model. Furthermore, we showed how to deploy the CLipSwish activation function. For both i-DenseNets and Residual Flows this significantly improves performance. Smaller architectures under an equal parameter budget were used for the exploration of different settings.

The full model for density estimation was trained on $32 \times 32$ CIFAR10 and ImageNet32 data. We demonstrated the performance of i-DenseNets and compared the models to Residual Flows and other comparable Flow-based models on density estimation in bits per dimension. Yet, it also demonstrated how the model could be deployed for hybrid modeling that includes classification in terms of accuracy and density estimation in bits per dimension. Furthermore, we showed that modeling ResNet connections matching the size of an i-DenseNet obtained worse performance than the i-DenseNet and the original Residual Flow. In conclusion, i-DenseNets out-perform Residual Flows and other competitive flow-based models for density estimation on all considered datasets in bits per dimension and hybrid modeling that includes classification. The obtained results clearly indicate the high potential of i-DenseNets as powerful flow-based models.

## Acknowledgments

We would like to thank Patrick Forré for his helpful feedback on the derivations. Furthermore, this work was carried out on the Dutch national e-infrastructure with the support of SURF Cooperative.

## Funding Transparency Statement

There are no additional sources of funding to disclose.

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
