# A Derivations

This appendix provides the reader full derivations of the Lipschitz constant for the concatenation in DenseNets and a bound of the Lipschitz for the activation functions.

## A.1 Derivation of Lipschitz constant K for the concatenation

We know that a function $f$ is K-Lipschitz if for all points $v$ and $w$ the following holds:

$$d_Y(f(v), f(w)) \leq K d_X(v, w), \tag{15}$$

where $d_Y$ and $d_X$ are distance metrics and $K$ is the Lipschitz constant.

Consider the case where we assume to have the same distance metric $d_Y = d_X = d$, and where the distance metric is assumed to be chosen as any $p$-norm, where $p \geq 1$, for vectors: $||\delta||_p = \sqrt[p]{\sum_{i=1}^{len(\delta)} |\delta_i|^p}$. Further, we assume a DenseBlock to be a function $h$ where the output for each data point $v$ and $w$ is expressed as follows:

$$h_v = \begin{bmatrix} f_1(v) \\ f_2(v) \end{bmatrix}, \quad h_w = \begin{bmatrix} f_1(w) \\ f_2(w) \end{bmatrix}, \tag{16}$$

where in this paper for a Dense Layer and for a data point $x$ the function $f_1(x) = x$ and $f_2$ expresses a linear combination of (convolutional) weights with $x$ followed by a non-linearity, for example $\phi(W_1 x)$. We can rewrite Equation (15) for the DenseNet function as:

$$d(h_v, h_w) \leq K d(v, w), \tag{17}$$

where K is the unknown Lipschitz constant for the entire DenseBlock. However, we can find an analytical form to express a limit on K. To solve this, we know that the distance between $h_v$ and $h_w$ can be expressed by the $p$-norm as:

$$d(h_v, h_w) = \sqrt[p]{\sum_{i=1}^{len(h_v)} |h_{v,i} - h_{w,i}|^p}, \tag{18}$$

where we can simplify the equation by taking the $p$-th power:

$$d(h_v, h_w)^p = \sum_{i=1}^{len(f_1(v))} |f_1(v)_i - f_1(w)_i|^p + \sum_{i=1}^{len(f_2(v))} |f_2(v)_i - f_2(w)_i|^p. \tag{19}$$

Since we know that the distance of $f_1$ can be expressed as:

$$d(f_1(v), f_1(w)) = \sqrt[p]{\sum_{i=1}^{len(f_1(v))} |f_1(v)_i - f_1(w)_i|^p}, \tag{20}$$

which is similar for the distance of $f_2$, re-writing the second term of Equation (19) in the form of Equation (17) is assumed to be of form:

$$d(f_1(v), f_1(w))^p \leq K_1^p d(v, w)^p, \tag{21}$$

which is similar for $f_2$, $d(f_2(v), f_2(w))^p \leq K_2^p d(v, w)^p$. Assuming this, we can find a form of Equation (17) by substituting Equation (19) and Equation (21):

$$d(h_v, h_w)^p = \sum_{i}^{len(h_v)} |h_{v,i} - h_{w,i}|^p \leq d(f_1(v), f_1(w))^p + d(f_2(v), f_2(w))^p$$
$$= (K_1^p + K_2^p) d(v, w)^p. \tag{22}$$

Now, taking the $p$-th root we have:

$$d(h_v, h_w) \leq \sqrt[p]{(\mathrm{K}_1^p + \mathrm{K}_2^p)} d(v, w), \tag{23}$$

where we have derived the form of Equation (17) and where $\mathrm{Lip}(h) = \mathrm{K}$ is expressed as:

$$\mathrm{Lip}(h) = \sqrt[p]{(\mathrm{K}_1^p + \mathrm{K}_2^p)}, \tag{24}$$

where $\mathrm{Lip}(f_1) = \mathrm{K}_1$ and $\mathrm{Lip}(f_2) = \mathrm{K}_2$, which are assumed to be known Lipschitz constants.

## A.2 Derivation bounded Lipschitz Concatenated ReLU

We define function $\phi : \mathbb{R} \to \mathbb{R}^2$ as the Concatenated ReLU for a point $x$:

$$\phi(x) = \begin{bmatrix} \mathrm{ReLU}(x) \\ \mathrm{ReLU}(-x) \end{bmatrix}. \tag{25}$$

Let points $v, w \in \mathbb{R}$. From Section A.1, Equation (18), we know that the distance between points transformed with $\phi$ and using the $\ell_2$-norm can be written as:

$$\begin{aligned}
d(\phi(v), \phi(w))^2 &= \sum_{i=1}^{len(\phi(v))} |\phi(v)_i - \phi(w)_i|^2 \\
&= (\phi(v)_1 - \phi(w)_1)^2 + (\phi(v)_2 - \phi(w)_2)^2 \\
&= (\mathrm{ReLU}(v) - \mathrm{ReLU}(w))^2 + (\mathrm{ReLU}(-v) - \mathrm{ReLU}(-w))^2.
\end{aligned} \tag{26}$$

Furthermore, we know that the distance between the two points is:

$$\begin{aligned}
d(v, w)^2 &= \sum_{i=1}^{len(v)} (v_i - w_i)^2 \\
&= (v - w)^2 \\
&= v^2 + w^2 - 2vw.
\end{aligned} \tag{27}$$

We have four different situations that can happen. If $v > 0, w > 0$, then the distance between the points will be:

$$\begin{aligned}
d(\phi(v), \phi(w))^2 &= (v - w)^2 + 0 \\
&= d(v, w)^2.
\end{aligned} \tag{28}$$

In this specific case we have that $d(v, w)^2 = v^2 + w^2 - 2vw$, where $2vw > 0$. The same holds for $v \leq 0, w \leq 0$, when the first term becomes zero and instead of zero, the second term becomes $d(v, w)^2$ with $2vw \geq 0$.

If $v > 0, w \leq 0$, the distance between the points is equal to:

$$\begin{aligned}
d(\phi(v), \phi(w))^2 &= (v - 0)^2 + (0 - w)^2 \\
&= v^2 + w^2 \\
&= (v - w)^2 + \underbrace{2vw}_{\leq 0} \leq (v - w)^2 = d(v, w)^2.
\end{aligned} \tag{29}$$

The same derivation holds in the case $v \leq 0, w > 0$. Combining all cases, we find that $d(\phi(v), \phi(w)) \leq d(v, w)$, therefore:

$$\mathrm{Lip}(\mathrm{CReLU}) = 1. \tag{30}$$

## A.3 Derivation Lipschitz bound of CLipSwish

We propose the Concatenated LipSwish (CLipSwish) and show how we can enforce the CLipSwish to be 1-Lipschitz for a 1-dimensional input signal $x$ and generalization to a higher dimension in the upcoming subsections.

### A.3.1 CLipSwish 1 dimensional input signal

We derive the upper bound of Concatenated LipSwish and show that $\text{CLipSwish}(x) = \Phi(x)/1.004$ is enforced to satisfy $\text{Lip}(\text{CLipSwish}) = 1$. To start with, we define function $\Phi : \mathbb{R} \to \mathbb{R}^2$ for a point $x$ as:

$$\Phi(x) = \begin{bmatrix} \phi_1(x) \\ \phi_2(x) \end{bmatrix} = \begin{bmatrix} \text{LipSwish}(x) \\ \text{LipSwish}(-x) \end{bmatrix}, \tag{31}$$

where:

$$\text{LipSwish}(x) = x\sigma(\beta x)/1.1,$$

and the partial derivative of $\Phi(x)$ exists. Then the Jacobian matrix of $\Phi$ is well-defined as:

$$J_\Phi(x) = \begin{bmatrix} \frac{\partial \phi_1(x)}{\partial x} \\ \frac{\partial \phi_2(x)}{\partial x} \end{bmatrix}. \tag{32}$$

Furthermore, we know that for a $\ell_2$-Lipschitz bounded function $\Phi$, the following holds:

$$\text{Lip}(\Phi) = \sup_x ||J_\Phi(x)||_2, \tag{33}$$

where $J_\Phi(x)$ is the Jacobian of $\Phi$ and norm and $||\cdot||_2$ represents the induced matrix norm which is equal to the spectral norm of the matrix. Furthermore, we know that for a matrix $A$ the following holds: $||A||_2 = \sigma_{max}(A)$, where $\sigma_{max}$ is the largest singular value and the largest singular value is given by $\sigma_{max}(A) = \sqrt{\lambda_1}$, since $\sigma_i = \sqrt{\lambda_i}$ for $i = 1, \ldots, n$ [23]. Now determining the singular values of $J_\Phi(x)$ is done by solving $\det(J_\Phi(x)^T J_\Phi(x) - \lambda I_n) = 0$. Combining and solving gives:

$$\det(J_\Phi(x)^T J_\Phi(x) - \lambda I_n) = 0$$
$$\left[ \left( \frac{\partial \phi_1(x)}{\partial x} \right)^2 + \left( \frac{\partial \phi_2(x)}{\partial x} \right)^2 \right] - \lambda = 0 \tag{34}$$
$$\lambda = \left( \frac{\partial \phi_1(x)}{\partial x} \right)^2 + \left( \frac{\partial \phi_2(x)}{\partial x} \right)^2$$

where $\lambda = \lambda_1$ the largest eigenvalue, thus: $\lambda_1 = \left( \frac{\partial \phi_1(x)}{\partial x} \right)^2 + \left( \frac{\partial \phi_2(x)}{\partial x} \right)^2$. Therefore, the spectral norm of Equation (33), can be re-written as:

$$||J_\Phi(x)||_2 = \sigma_{max}(J_\Phi(x)) = \sqrt{\left( \frac{\partial \phi_1(x)}{\partial x} \right)^2 + \left( \frac{\partial \phi_2(x)}{\partial x} \right)^2}. \tag{35}$$

Now $\text{Lip}(\Phi)$ is the upper bound of the CLipSwish and is equal to the supremum of: $\text{Lip}(\Phi) = \sup_x ||J_\Phi(x)||_2 \approx 1.004$, for all values of $\beta$. This can be numerically computed by any solver, by determining the extreme values of Equation (35).

### A.3.2 Generalization to higher dimensions

To generalize the Concatenated LipSwish activation function activation function to higher dimensions, we take $\Phi : \mathbb{R}^d \to \mathbb{R}^{2d}$, which represents the CLipSwish activation function for a vector $\mathbf{x} = \{x_1, x_2, \ldots, x_d\}$. Then the CLipSwish is given by the concatenation $\Phi(\mathbf{x}) = [\text{LipSwish}(\mathbf{x}), \quad \text{LipSwish}(-\mathbf{x})]$, where $\phi_1(\mathbf{x}) = \text{LipSwish}(\mathbf{x})$ and $\phi_2(\mathbf{x}) = \text{LipSwish}(-\mathbf{x})$ ele-

mentwise. The Jacobian matrix $J_\Phi(\mathbf{x})$ with shape $2d \times d$, looks as follows:

$$J_\Phi(\mathbf{x}) = \begin{bmatrix} \frac{\partial \phi_1(\mathbf{x})_1}{\partial x_1} & \frac{\partial \phi_1(\mathbf{x})_1}{\partial x_2} & \cdots & \frac{\partial \phi_1(\mathbf{x})_1}{\partial x_d} \\ \vdots & \vdots & & \vdots \\ \frac{\partial \phi_1(\mathbf{x})_d}{\partial x_1} & \frac{\partial \phi_1(\mathbf{x})_d}{\partial x_2} & \cdots & \frac{\partial \phi_1(\mathbf{x})_d}{\partial x_d} \\ \frac{\partial \phi_2(\mathbf{x})_1}{\partial x_1} & \frac{\partial \phi_2(\mathbf{x})_1}{\partial x_2} & \cdots & \frac{\partial \phi_2(\mathbf{x})_1}{\partial x_d} \\ \vdots & \vdots & & \vdots \\ \frac{\partial \phi_2(\mathbf{x})_d}{\partial x_1} & \frac{\partial \phi_2(\mathbf{x})_d}{\partial x_2} & \cdots & \frac{\partial \phi_2(\mathbf{x})_d}{\partial x_d} \end{bmatrix}, \tag{36}$$

where $\frac{\partial \phi_{i,j}}{\partial x_k} = \begin{cases} 0, & \text{if } j \neq k \\ \frac{\partial \phi_{i,j}}{\partial x_k}, & \text{otherwise.} \end{cases}$

The determinant is computed as $\det(J_\Phi(\mathbf{x})^T J_\Phi(\mathbf{x}) - \lambda I_n) = 0$, where $J_\Phi(\mathbf{x})^T J_\Phi(\mathbf{x})$ is of shape $d \times d$ with off-diagonals equal to zero. Therefore, the determinant is given by multiplication of the diagonal entries and each eigenvalue is given by each diagonal entry. The general form of determinant and eigenvalues is written as:

$$\det(J_\Phi(\mathbf{x})^T J_\Phi(\mathbf{x}) - \lambda I_n) = \prod_{j=1}^{d} \lambda_j, \tag{37}$$

where each eigenvalue is given by:

$$\lambda_j = \left( \frac{\partial \phi_{1,j}}{\partial x_j} \right)^2 + \left( \frac{\partial \phi_{2,j}}{\partial x_j} \right)^2. \tag{38}$$

Then:

$$\text{Lip}(\Phi) = \sup_x ||J_\Phi(x)||_2 = \sup_x \max_j \sqrt{\lambda_j} = \sup_x \max_j \sqrt{\left( \frac{\partial \phi_{1,j}}{\partial x_j} \right)^2 + \left( \frac{\partial \phi_{2,j}}{\partial x_j} \right)^2} \approx 1.004, \tag{39}$$

where the last step is numerically approximated for the CLipSwish function, where $\phi$ is the LipSwish. Therefore, we plot Equation (39) in Python and compute the absolute extrema, which can be found in Figure 6. For this figure we plotted CLipSwish with $\beta = 0.5$ and passed it through a softplus function, as it is initialized in the code on GitHub. Next, we can numerically obtain the absolute extrema by computing the maximum value and argmax of the maximum value of Equation (39), which respectively represent the y-coordinate and x-coordinate of the absolute maximum. This accounts for all $\beta$'s being strictly positive since changing $\beta$ does not change the y-coordinate of the extreme value but only shifts the x-coordinate more to or further away from the origin.

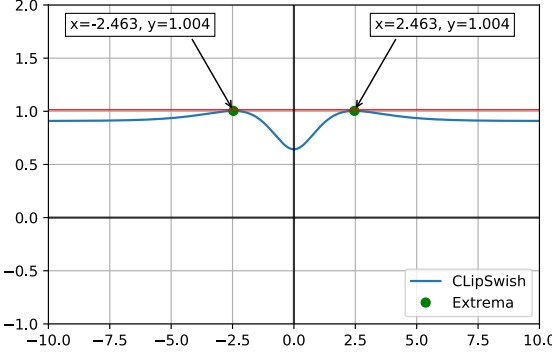

Figure 6: ClipSwish activation function with indicated absolute maximums

# B Implementation

We followed the architecture of [4] and during training used a batch size of 64. CIFAR10 and ImageNet32 are of size $32 \times 32$. CIFAR10 contains 50,000 training images and 10,000 test images. ImageNet32 contains 1,281,167 training images and 50,000 validation images. CIFAR10 has an MIT License and the ImageNet terms of access can be found here: `https://image-net.org/download.php`. Before training, uniform dequantization is applied to the images after which a logit transformation is applied. For hybrid models, instead of the logit transform, the images use normalization $x = \frac{x-\mu}{\sigma}$. As in [4], for evaluation at least 20 terms of the power series for the Jacobian-determinant are computed while the remaining terms to compute, are determined by the unbiased estimator. Furthermore, we set a bound on the Lipschitz constant of each dense layer with a Lipschitz coefficient of $0.98$. We use Adam optimizer with learning rate set to $0.001$ to train the models.

For all our models we ensured an equal parameter budget as the architecture of Residual Flows [4]. For CIFAR10, the full i-DenseNets utilize 24.9M to utilize the 25.2M of Residual Flows. For ImageNet32, i-DenseNet utilizes 47.0M parameters to utilize the 47.1M of the Residual Flow. A numerical architecture of the full i-DenseNets for image data is presented in Table 7. $g$ consists of several dense layers. The last dense layer $h_n$ is followed by a $1 \times 1$ convolution to match the output of size $\mathbb{R}^d$, after which a squeezing layer is applied. The final part of the network consists of a Fully Connected (FC) layer with the number of blocks set to 4 for both datasets. Before the concatenation in the FC layer, a Linear layer of input $\mathbb{R}^d$ to output dimension $64$ is applied, followed by the dense layer with for both datasets the FC DenseNet growth of 32, activation CLipSwish and a DenseNet depth of 3. The final part consists of a Linear layer to match the output of size $\mathbb{R}^d$. The large-scale models require approximately 410 seconds for an epoch on 4 NVIDIA TITAN RTX GPUs.

Table 7: The general DenseNet architecture for the full models, modeled in function $g$ for image data.

| Nr. of scales | Nr. of blocks per scale | DenseNet Depth | DenseNet Growth | Dense Layer | Output |
|---|---|---|---|---|---|
| 3 | 16 (CIFAR10) 32 (ImageNet32) | 3 | 172 (CIFAR10) 172 (ImageNet32) | $\begin{bmatrix} 3 \times 3 & \text{conv} \\ \text{CLipSwish} \\ \text{concat} \end{bmatrix}$ | $[1 \times 1 \quad \text{conv}]$ |

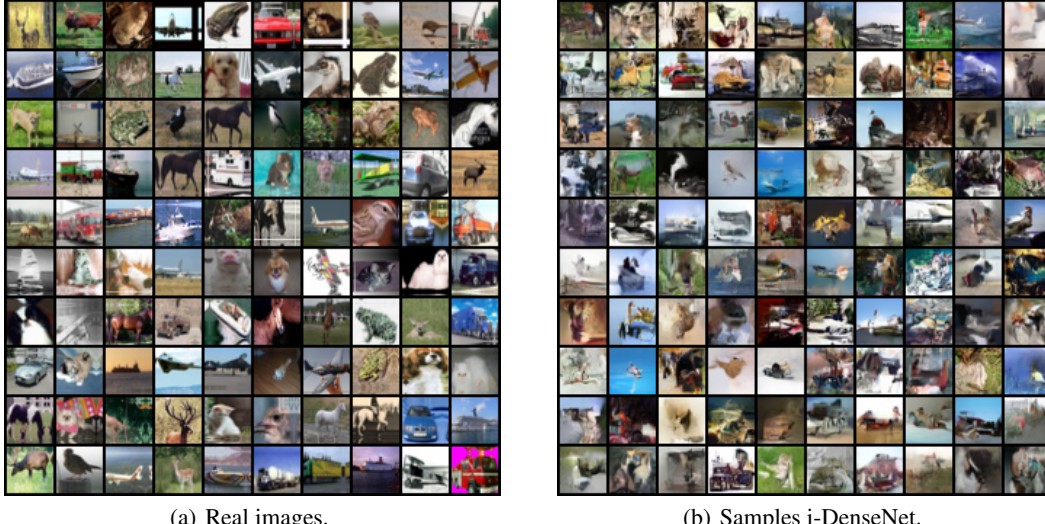

(a) Real images.

(b) Samples i-DenseNet.

Figure 7: Real images and samples from i-DenseNet trained on CIFAR10.

# C    Samples

This appendix contains samples of the models trained on CIFAR10 and ImageNet32, along with samples of the hybrid models.

## C.1    Model Samples

Figure 7 shows real images and samples of the models trained on CIFAR10. Figure 7(a) shows the real images and Figure 7(b) shows samples of i-DenseNet trained on CIFAR10.

Figure 8 contains real images and samples of the models trained on ImageNet32. Figure 8(a) shows the real images and Figure 8(b) shows samples of i-DenseNet trained on ImageNet32.

## C.2    Hybrid modeling samples

Figure 9 shows samples of the hybrid models trained on CIFAR10. The model trained with a scaling factor of $\lambda = \frac{1}{D}$ can be found in Figure 9(a). We notice that the samples tend to show a lot of red and brown colors and that the images tend to look noisy. This is probably due to the scaling factor where the generative part and classifier part have an equal focus for the likelihood objective, while there are $D = 32 \times 32$ features per image.

The model trained with $\lambda = 1$ can be found in Figure 9(b). The samples tend to look like the samples in Figure 7(b), only with less definition. This is probably due to the extra part, namely, the classifier part. Comparing the bits per dimension of the hybrid model with i-DenseNet trained for density estimation only, we find a difference of $0.06$bpd.

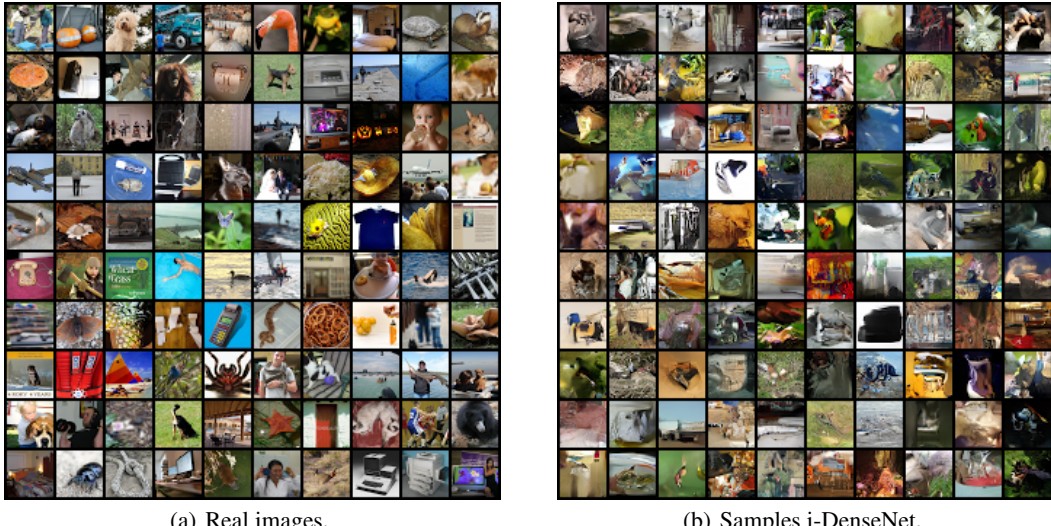

(a) Real images.

(b) Samples i-DenseNet.

Figure 8: Real images and samples from i-DenseNet trained on ImageNet32.

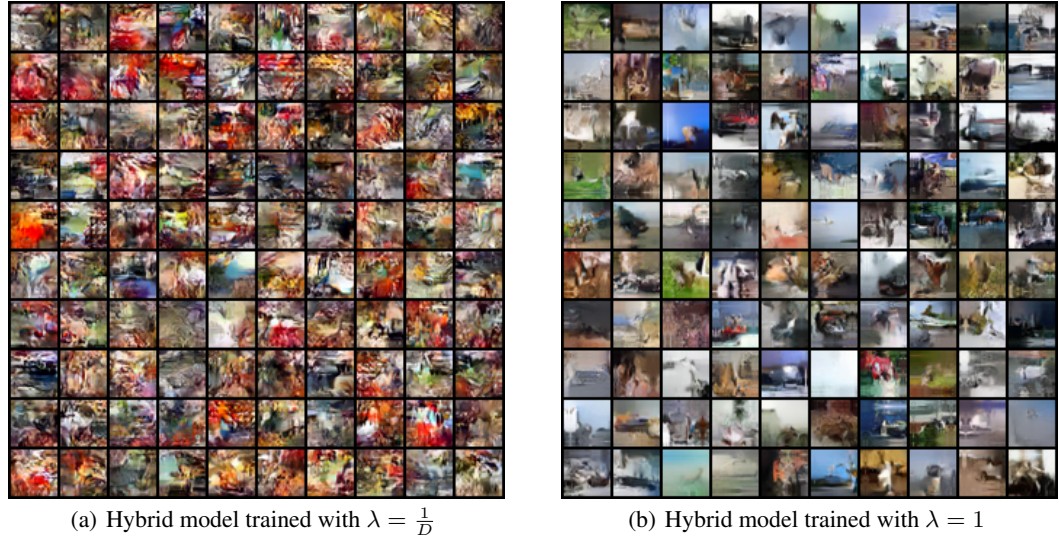

(a) Hybrid model trained with $\lambda = \frac{1}{D}$

(b) Hybrid model trained with $\lambda = 1$

Figure 9: Hybrid modeling results with Dense Blocks trained on CIFAR10

# D   Additional experiments

In this appendix, we perform additional experiments. First, we analyze the preservation of signal for the activations functions with datapoints that are sampled from a distribution with larger parameter values. Furthermore, we analyze the running time of the models. Next, we examine the importance of the concatenated representation for i-DenseNets that are learned with learnable weighted concatenation. Finally, we analyze a Residual Flow where we extend the width and depth of the ResNet connections modeled in $g(x)$ such that it matches the size of i-DenseNet.

## D.1   Preservation of signal

To further analyze the expressive power for the activation functions with a larger range, we sample 100,000 datapoints from distribution: $v, w \sim \mathcal{N}(0, 5)$ with dimension set to $D = \{1, 128, 1024\}$. We compute the mean and maximum of the sampled ratios with: $\ell_2(\phi(v), \phi(w))/\ell_2(v, w)$ and analyze the expressive power of each function. Table 8 shows the results. We find that CLipSwish for all dimensions preserves even more expressive power than datapoints sampled from $\mathcal{N}(0, 1)$, while sigmoid loses a considerable amount of signal with mean values close to zero instead of $0.25$.

Table 8: The mean and maximum ratio for different dimensions with sample size set to 100,000.

| Activation\ Measure | $D = 1$ | | $D = 128$ | | $D = 1024$ | |
|---|---|---|---|---|---|---|
| | Mean | Max | Mean | Max | Mean | Max |
| Sigmoid | 0.09 | 0.25 | 0.08 | 0.10 | 0.08 | 0.09 |
| LipSwish | 0.47 | 1.0 | 0.54 | 0.69 | 0.54 | 0.59 |
| CLipSwish | 0.83 | 1.0 | 0.76 | 0.83 | 0.76 | 0.78 |
| Identity | 1.0 | 1.0 | 1.0 | 1.0 | 1.0 | 1.0 |

## D.2   Running time

Table 9 shows the forward pass, train time and sampling time, expressed in percentage faster or slower than Residual Flow, for each activation function. We find that the forward pass of i-DenseNet, for all activation functions, is faster than Residual Flow. The train time is slower and during sampling i-DenseNet is faster. Note that in comparison to the preliminary results in the rebuttal the times has changed somewhat, since these results have been obtained on a clean system with multiple runs. An interesting observation is that the LeakyLSwish with Residual Flows is much slower than the DenseNet variant, which indicates that fewer fixed-point iterations are needed for i-DenseNets to converge.

Table 9: i-DenseNet approximate running times in percentage (%) compared to Residual Flow. Faster than Residual Flow is indicated with ↑ and slower ↓.

| Activation Function | Forward pass (GPU) | Train time (GPU) | Sampling time (CPU) |
|---|---|---|---|
| LipSwish | ↑ 1.3% | ↓ 43% | ↑ 8.8% |
| LeakyLSwish | ↑ 1.3% | ↓ 28% | ↑ 231.9% |
| CLipSwish | ↑ 0.6% | ↓ 145% | ↑ 11% |

## D.3   Importance of concatenated representation

Trained on CIFAR10, the smaller architecture with CLipSwish activation and a DenseNet depth of 3 and growth of 178, run for 200 epochs with CLipSwish obtains the best performance score with 3.37bpd. To analyze the importance of the concatenated representation after training, Figure 10 shows the heatmap for parameter $\hat{\eta}_1$ (Figure 10(a)) and parameter $\hat{\eta}_2$ (Figure 10(b)). Every scale level 1, 2, and 3 contains 4 DenseBlocks, that each contains 3 dense layers with convolutional layers.

The final level FC indicates that fully connected layers are used. The letters 'a', 'b', and 'c' index the dense layers per block.

Remarkably, all scale levels for the last layers $h_{ic}$ give little importance to the input signal. The input signals for these layers are in most cases multiplied with $\hat{\eta}_1$ (close to) zero, while the transformed signal uses almost all the information when multiplied with $\hat{\eta}_2$, which is close to one. This indicates that the transformed signal is of more importance for the network than the input signal. For the fully connected part, this difference is not that pronounced. Instead of 4 DenseBlocks, the full i-DenseNet model utilizes 16 DenseBlocks (CIFAR10) and 32 (ImageNet32) for every scale; these are not included due to the size.

## D.4 Matching architectures

The Residual Flow architecture with LipSwish activation and 3 scale levels set to 4 Flow blocks has 8.7M parameters. To utilize a similar number of parameters for i-DenseNet with LipSwish activation, we set DenseNets depth to 3 and growth to 124. To go a step further, we also examine modeling ResNet connections matching the size of i-DenseNet. Therefore, we use the same $3 \times 3$ kernels as each dense layer uses and as a final layer a $1 \times 1$ kernel to match the input size. Instead of the concatenation, we use the growth size of 124 plus the input size to imitate the dense layers of i-DenseNet but then with convolutional connections. We repeat this process for the Fully Connected layer. Note that this puts the Residual Flow at a considerable advantage as it uses 19.1M parameters instead of the 8.7M of the original flow. We do the same experiment for toy data that uses only linear connections instead of convolutions.

In Table 10 the results are shown. On toy data, the extended Residual Flow performs slightly better in terms of nats compared to the original Residual Flow without extended width and depth. Yet, i-DenseNet obtains the lowest (better) scores. On high-dimensional CIFAR10 data, the extended Residual Flow obtains 7.02bpd which is worse than i-DenseNet with 3.39bpd. Yet, the model also scores more than double as high (worse) in terms of bpd than the original Residual Flow with 3.42bpd.

Table 10: The negative log-likelihood results on test data in nats (toy data) and bpd (CIFAR10), where ↓ lower is better. i-DenseNets with LC are compared with the original Residual Flow and Residual Flow with equal width and depth as i-DenseNet.

| Model (LipSwish) | CIFAR10 *bpd ↓* | 2 circles *nats ↓* | checkerboard *nats ↓* | 2 moons *nats ↓* |
|---|---|---|---|---|
| Residual Flows | 3.42 | 3.44 | 3.81 | 2.60 |
| + extended width, equal depth | 7.02 | 3.36 | 3.78 | 2.52 |
| **i-DenseNets+LC** | **3.39** | **3.30** | **3.66** | **2.39** |

The main difference in architecture of the toy and CIFAR10 is the linear layer for toy data whereas mainly convolutional layers are used for CIFAR10. A possible explanation of this phenomenon is that by forcing more convolutional layers to be 1-Lipschitz that *gradient norm attenuation* problems increase and in practice become less expressive. Concluding, even though the model utilizes more than double the number of parameters, it performs worse than i-DenseNet with similar architecture and even worse than the original Residual Flow architecture, indicating that modeling a DenseNet in $g(x)$ indeed is an important difference that gives better performance.

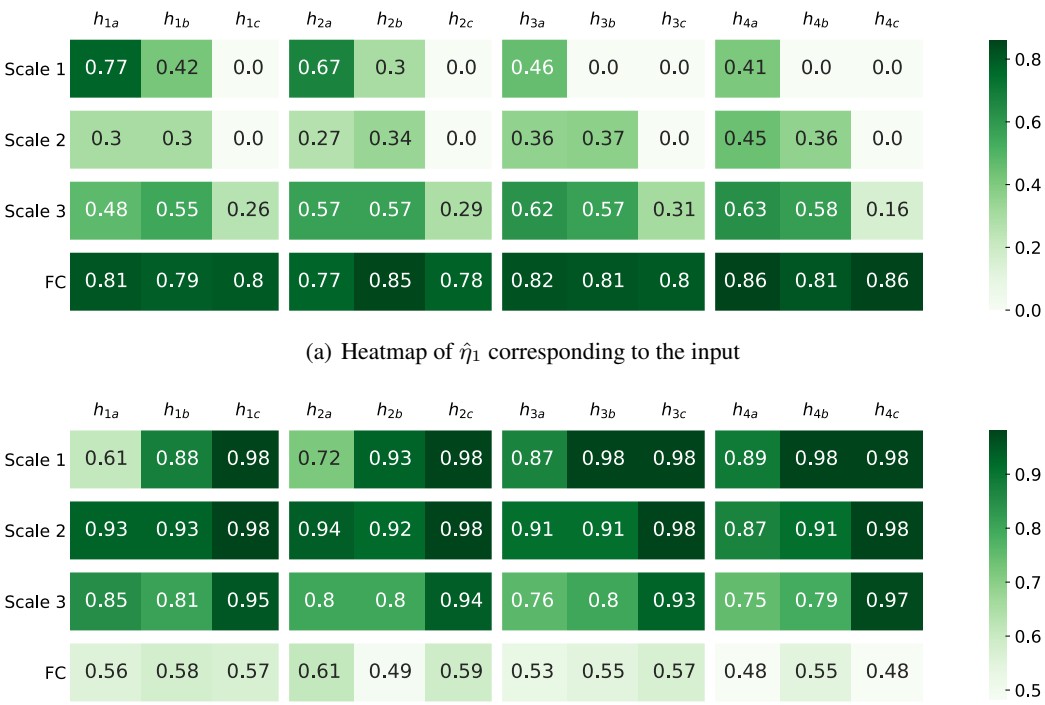

(a) Heatmap of $\hat{\eta}_1$ corresponding to the input

(b) Heatmap of $\hat{\eta}_2$ corresponding to the transformed input

Figure 10: Heatmaps of the normalized $\eta_1$ and $\eta_2$ after training for 200 epochs on CIFAR10. Best viewed electronically.