# OpenReview forum: "Invertible DenseNets with Concatenated LipSwish"
_NeurIPS.cc/2021/Conference — NeurIPS 2021 Poster_

### Official Review · Reviewer_rhhy · 2021-07-14

**Rating:** 7
**Confidence:** 4

**Summary:**

In the article, a new normalizing flow architecture, called invertible DenseNets (i-DenseNets), is introduced. It is an extension of residual flows with two new contributions. First, the invertible residual blocks are replaced by invertible dense blocks. Similar to residual blocks, the authors proof invertibility by computing the Lipschitz constant and guarantee that it is smaller than one by a suitable rescaling operation. Second, the LipSwish activation function is replaced by its concatenated cousin, which is rescaled as well to ensure invertibility.
The method is applied to three illustrative toy datasets as well as CIFAR10 and ImageNet32, where it outperforms competing methods and sets a new SOTA for flow models with uniform dequantization. They also use an i-DenseNet to perform hybrid modelling, where is outperform the competition as well.

**Limitations And Societal Impact:**

The main limitation is that the model has not been applied to high resolution data and the computational cost required to train it is unclear.
The societal impact regarding the generation of misinformation is discussed in the paper.

**Main Review:**

DenseNets are an important class of neural networks and modifying them to be invertible is an important contribution to machine learning. The authors describe their model very clearly and illustrate the architecture well. In secion 5, they also give interesting insights in why the individual parts of their model are important and lead a better performance in the experiments. However, their style to organize their argumentation is unusual. I would recommend moving some of the arguments made in section 5.1 to 5.3 to section 3 and show the respective experimental results at the beginning of section 4. Thereby, the reader would directly get a better understanding of the reason for the architecture choice.
Regarding the experiments, the reported results are impressive. However, it would be interesting to apply this method to higher dimensional datasets as it is done in most articles that introduced the competing methods discussed in this paper. Also, running each experiments with multiple seeds would be desirable, although I acknowledge that this is often not the case in similar published articles. Also, the computational cost of training i-DenseNets especially compared to residual flows is not discussed, which is unfortunate since the fact that DenseNets require less computation than ResNets is stated as one main motivation of this work.

**Time Spent Reviewing:**

3

---

> ### Author Response · Authors · 2021-08-09
> **Response to rhhy**
>
> Thank you rhhy for your comments. The reviewer thinks our method is an important contribution to the community.
>
> In addition, the reviewer suggests subdividing section 5 under section 3 and 4 for readability and better understanding of the architecture choices at an earlier stage. We see the perspective of this view. For this reason, we will move some of the considerations to the earlier sections, so that a reader is aware of them at an earlier point.
>
> Further, about the computational cost, we agree that a running time analysis is useful. An initial experiment showed that for the models used in Table 6, the forward pass of i-DenseNets with LipSwish is roughly 1% faster (to compute the likelihood) while the backward pass is 40% slower than the Residual Flow. Part of this difference can be explained by the concatenation operation for which feature maps need to be copied in this framework, which puts i-DenseNets at a slight disadvantage for this specific implementation. During sampling i-Densenet is roughly 45% faster than Residual Flow. We will include an analysis in the updated version of the paper.
>
> We hope this has clarified your questions. Feel free to reach out if you have any further comments.

---

> > ### Comment · Reviewer_rhhy · 2021-08-25
> > **Reply to Paper1446 Authors**
> >
> > Thank you for the reply. I appreciate that the authors will follow my suggestion and restructure section 5. I still think this is a good paper, which is worth being accepted.

---

### Official Review · Reviewer_n9rc · 2021-07-14

**Rating:** 7
**Confidence:** 3

**Summary:**

The paper proposes a parameter-efficient DenseNet block and a CLipSwish (concatenated LipSwish) activation function for use with the DenseNet blocks in the residual flows setting.  The paper outperforms prior art (with uniform dequantization) in density estimation. Results are reported on CIFAR10 and ImageNet32 with rich ablation studies.

**Limitations And Societal Impact:**

Appropriately discussed in the "Societal Impact" section.

**Main Review:**

The idea of using densenet blocks and constraining its Lipschitz constant in the residual flow setting is interesting and novel. The paper is clearly written and easy to follow, with a few exceptions listed below. The work is quite self-contained and seems not to have loose ends. The main motivation for adopting densenet blocks is improving parameter efficiency and density estimation performance; both goals have been achieved according to the experiments.

A few questions and remarks for improvement:

- Fig.1 violet part is not clear
- Eq.2 uses W to denote matrix (norm), while Fig.1 uses W to denote (convolutional) layers. While this is in line with the prior art (Invertible ResNet), perhaps the notation could be adjusted for clarity.
- L76 missing subscript _2 of the norm
- L93 could benefit from explaining how this is the same as Eq.2 from the DenseNet paper ($x_l = H_l([x_0,...,x_{l-1}]$)
- L117,122 usage of 'dividing' is not precise
- L134: missing {} around etas
- L141 missing explanation for the constant
- L162 in relation to Appendix:L584-586 could elaborate on the computation of the exact value (perhaps with a plot?)
- L169 channel growth, as a term related to densenet blocks, perhaps could be mentioned earlier
- L186 refers to [4] for baselines using uniform dequantization, but to introduce a reader to types of dequantizaton, a better reference is [13]
- Sec.4.2 (in relation to CIFAR-10 and ImageNet32 experiments): how would the proposed method score in perceptual quality metrics, such as FID?
- Sec.5.1 while the analysis of individual activation functions is interesting, it would also be interesting to evaluate the whole model using e.g., a histogram of the empirical Lipschitz constant (eLhist) [1]

[1] Sanyal et al., Stable rank normalization for improved generalization in neural networks and GANs

============== Post-rebuttal comment

After reading all reviews and responses, I decided to keep my original rating (A).

**Time Spent Reviewing:**

4

---

> ### Author Response · Authors · 2021-08-09
> **Response to n9rc**
>
> Thank you n9rc for your questions and remarks for improvement. To clarify your remarks:
>
> - The violet part was used to show the 1x1 convolution to reduce the dimension of the last dense layer to match the output shape. We will add a short clarification in the figure caption.
> - Since a convolutional layer can be expressed as a matrix multiplication, we opted for the general notation of W for both fully connected layers and convolutional layers. We agree with the reviewer that this should be clear and we will add a remark about this notational choice.
> - We will give an explanation of the constant in L141 and add a plot for Eq.35 in the Appendix that presents the graphical structure of the activation function. We will add [13] as a reference for other dequantization types.
> - For the remaining remarks and comments, we appreciate these and will update them in the paper accordingly.
>
> We hope this has clarified your questions. Feel free to reach out if you have any further comments.

---

### Official Review · Reviewer_psZ5 · 2021-07-16

**Rating:** 8
**Confidence:** 4

**Summary:**

This paper proposes to use a DenseNet as the residual function of Residual Flows. The paper carefully describes how to ensure the Lipschitz constant is bounded when performing the concatenation of representations inherent in DenseNets. The paper also introduces a new activation function called Concatenated LipSwish that may be of independent interest. Finally, empirical results show that the proposed i-DenseNets perform better than Residual Flows and other baselines.


**Limitations And Societal Impact:**

- Reasonable discussion of possible negative societal impact.


**Main Review:**

**Summary of review**
Overall, this is a clearly written paper that proposes a novel model architecture, provides important insights into the choices and architecture, and gives good empirical results.

**Strengths:**
- Very clear discussion of prior works and core contributions including the learnable weight concatenation.

- Proposed a careful incorporation of DenseNets as residual function fo Residual Flows where the Lipschitz constant can be controlled.

- Proposed a new trainable activation function called concatenated LipSwish that preserves more information flow.

- Outperforms standard baselines including Residual Flows and Flow++ when using uniform dequantization.

- Interesting and useful analysis of activation functions, concatenation depth, and DenseNets vs ResNets to practically understand some key hyperparameters and architectural constraints.

**Weaknesses:**
- The actual empirical benefits over Residual Flows seems small (though it does seem to improve in all cases) so it is not clear that i-DenseNets should always be chosen over Residual Flows.

- Related to above, no training time comparison was given.  Could you give wall-clock training times for Residual Flows vs i-DenseNets? This may be a key factor in determining which one to use in practice.  Also, could you provide a comparison for time for inference at test time?

- The novelty is medium.  The analysis of CLipSwish is very good and may become a standard.  However, the other ideas are mainly careful synthesis of other ideas.

**Other comments or questions**
- I'd suggest emphasizing the difference of $h{n+1}$ more.  Maybe use a different symbol since it is very different then the other $h$'s.

- I did not understand "learnable weight concatenation" until the section that gives the details.  Maybe try to describe that it is related to making the Lipschitz constant less than 1 for the dense layer.

- It may be helpful to mention immediately that the CLipSwish function doubles the channel number. This is mentioned at the end of the section but having it earlier may help understanding.

**Typos or other minor comments**
- L320 "boost performs" -> "boosts performance"



**Time Spent Reviewing:**

1.25

---

> ### Author Response · Authors · 2021-08-09
> **Response to psZ5**
>
> Thank you psZ5 for your remarks. To clarify your comments.
>
> On choosing i-DenseNets over Residual Flows: Please note that although the performance differences for density modelling are quite close, the benefits of i-DenseNets for hybrid modelling are far more pronounced. For instance for lambda 1/D: i-DenseNets improve 90.8% accuracy with 3.49 bpd, versus 90.5% with only 3.62 bpd, a difference of 0.13 bpd (Table 4). This suggests that for hybrid modeling tasks, i-DenseNets have a clear advantage.
>
> Although we did provide running time for the full i-DenseNets, we agree that a running time analysis is useful. An initial experiment showed that for the models used in Table 6, the forward pass of i-DenseNets with LipSwish is roughly 1% faster while the backward pass is 40% slower than the Residual Flow. During sampling i-Densenet is roughly 45% faster than Residual Flow. We will include an analysis of this in an updated version of the paper.
>
> Regarding your other comments, thank you for pointing them out. We will emphasize the difference of the 1x1 conv by using another symbol and clarify the relation between the Lipschitz constant restriction for dense layers for the learnable weighted concatenation and mention that CLipSwish functions double the channel numbers earlier.
>
> We hope this has clarified your questions. Feel free to reach out if you have any further comments.

---

### Official Review · Reviewer_Lrfi · 2021-07-17

**Rating:** 6
**Confidence:** 4

**Summary:**

The paper extends Residual Flows from ResNet to DenseNet. In addition, the authors further improve the performance by modifying the Lipschitz activation.

**Limitations And Societal Impact:**

The paper discusses the limitations and potential societal impact in Section 5.4.

**Main Review:**

The paper is well written, and the presentation is clear. However, I feel the contribution is incremental, and the performance gain is marginal.

1. Since DenseNet has the same residual form F(x) = x + g(x), the authors inherit all important components (fixed-point iteration, Skilling-Hutchinson trace estimator, and "Russian roulette" strategy) from Residual Flow. Therefore, the extension to DenseNet is not complicated, which relies on the Lipscitzness of convex combination. In addition, the CLipSwish only modifies the original LipSwich activation slightly.

2. Given the marginal performance gain (e.g., Table 3), it is beneficial to report both the memory and actual running time for both Residual Flow and i-DenseNet. Due to the sequential structure of DenseNet, I wonder if the DenseNet with the same parameters is more expensive than Residual Flow in practice.

**Time Spent Reviewing:**

5

---

> ### Author Response · Authors · 2021-08-09
> **Response to Lrfi**
>
> Thank you Lrfi for the comments. The reviewer thinks the paper is well-written and the presentation is clear. To clarify the remarks:
>
> We agree that the CLipSwish is a tweak of the LipSwish activation function. However, we would like to add that the derivation for the concatenated version of the LipSwish activation relies on more than only convex combinations. Thereby, we provide a method to derive a Lipschitz bound for concatenated activation functions which could be beneficial in practice for other types of research that use Lipschitz constrained networks.
>
> Although we did provide running time for the full i-DenseNets, we agree that a running time analysis is useful. An initial experiment showed that for the models used in Table 6, the forward pass of i-DenseNets with LipSwish is roughly 1% faster while the backward pass is 40% slower than the Residual Flow. During sampling i-Densenet is roughly 45% faster than Residual Flow. We will include an analysis of this in an updated version of the paper.
>
> We hope this has clarified your questions. Feel free to reach out if you have any further comments.

---

> > ### Comment · Reviewer_Lrfi · 2021-08-30
> > **Reply to Author Response**
> >
> > I appreciate that the author adds a runtime comparison against Residual Flows. Thanks for your clarification.

---

### Decision · Program_Chairs · 2021-09-27

**Decision:**

Accept (Poster)

**Comment:**

This paper proposes an extension of Residual Flows to construct invertible deep neural nets. The paper features two technical contributions: The invertible residual blocks are replaced by invertible dense blocks, and the LipSwish activation function is replaced by a concatenated version. While these contributions can be considered marginal/incremental (reviewer Lrfi) and with medium novelty (reviewer psZ5), all reviewers agree the contributions offer enough value for being accepted.

The reviewers acknowledged the author response and they engaged in committee discussions. No serious concerns surfaced during the review process, and the reviewers reached a consensus recommendation to accept the paper.